# OpenReview forum: "Towards Comprehensive and Efficient Post Safety Alignment of Large Language Models via Safety Patching"
_ICLR.cc/2025/Conference — ICLR 2025 Conference Withdrawn Submission_

### Official Review · Reviewer_1JaV · 2024-10-28

**Soundness:** 3
**Presentation:** 2
**Contribution:** 2
**Rating:** 5
**Confidence:** 3

**Summary:**

This paper introduces a SafePatching framework to improve the safety of large language models (LLMs) while maintaining their utility. The major contribution is two types of safety patches.
- The safety enhancement patch utilizes gradient ascent on harmful data to train the model to avoid generating unsafe responses. It effectively helps the model "unlearn" unsafe behaviors by adjusting the model parameters to minimize the risk of producing harmful content.
- The over-safety mitigation patch, developed through gradient descent, is designed to prevent the model from being overly cautious. It fine-tunes the model to ensure it does not overly restrict or reject benign inputs that might superficially appear sensitive or risky.

 The approach is tested across multiple LLMs, showing better performance in reducing harmful outputs, handling over-safety, and preserving utility compared to several existing methods.

**Strengths:**

+ The proposed approach is easy to understand, logical, and appears to be effective.

+ It addresses a significant and timely problem.

+ The paper is overall well-written.

+ The unlearning and fine-tuning techniques used in SafePatch are not new, the originality comes from considering dual patching at the same time.

+ The paper includes an extensive set of experiments.

**Weaknesses:**

- Limited Novelty in Core Techniques

While the dual-patching approach is innovative in combining safety enhancement with over-safety mitigation, the core methods (e.g., gradient ascent and descent on harmful data) rely heavily on existing unlearning and fine-tuning techniques.

- Clarity on Practical Deployment

The paper would benefit from more actionable details regarding the real-world deployment of SafePatching, especially the requirements on the harmful data set.

-  Stability of SafePatching Approach

SafePatching's dual-patch integration requires careful parameter tuning, especially with the two gradient-based patches potentially introducing conflicts within the model. The process of managing these interactions, although effective, may lack robustness or generalizability across different architectures or types of prompts.

**Questions:**

In the SafePatching framework, Eq (1) and Eq (2) are designed to achieve two opposing objective by applying gradient-based updates in opposite directions on the same harmful dataset. Could you please clarify and elaborate how they are implemented given a harmful dataset?


In SafePatching, what requirements should a harmful dataset fulfill? For example, are there specific expectations concerning its size, diversity, or other characteristics? Additionally, are these requirements realistic for SafePatching's application in real-world scenarios?

**Details Of Ethics Concerns:**

- Potential for Misuse and Safety Bypass

The SafePatching framework’s dual-patch approach is designed to mitigate over-safety, allowing the model to respond to benign prompts with sensitive keywords. However, this opens up a risk of misuse if bad actors attempt to exploit this flexibility to bypass safety mechanisms deliberately.

---

> ### Author Response · Authors · 2024-11-19
> **Rebuttal by Authors (1/2)**
>
> We appreciate your positive feedback on the effectiveness and clarity of our SafePatching framework. Specifically, we are glad that you acknowledge the logical approach, timeliness of addressing a pressing safety issue, and the substantial experimental validation. We also appreciate the recognition that our approach to dual patching is innovative, as this simultaneous handling of both safety enhancement and over-safety mitigation represents a novel and impactful angle for LLM safety.
>
> ---
>
> > Weakness 1: While the dual-patching approach is innovative in combining safety enhancement with over-safety mitigation, the core methods (e.g., gradient ascent and descent on harmful data) rely heavily on existing unlearning and fine-tuning techniques.
>
> As you noted, the dual-patching approach in SafePatching is indeed our most innovative contribution and serves as the core of the framework. This approach is specifically designed to balance safety, over-safety, and general utility, achieving the best results across these dimensions, as evidenced by our experiments.
>
> In contrast, the goal of SafePatching is not to propose new unlearning or fine-tuning techniques, as this lies outside the scope of our study. The elegance of such patch derivation process is that we leverage **a single harmful dataset to generate both types of patches**, thereby avoiding additional data overhead. This efficient use of resources may enhance the framework’s practical utility in real-world applications.
>
> ---
>
> > Weakness 2 and Question 2: What requirements should a harmful dataset fulfill? Are these requirements realistic for SafePatching's application in real-world scenarios?
>
> The requirements for the harmful dataset in SafePatching are relatively **flexible**. As noted in `lines 975 – 978` of our paper, SafePatching is trained on harmful data derived from inputs that have successfully bypassed the model’s defenses.
>
> Obtaining such harmful data is feasible in real-world applications, as:
>
> - Even safety-aligned models can be prompted to produce harmful outputs [1].
>
> - Advanced automated red-teaming techniques are also now capable of efficiently identifying model vulnerabilities [2,3], helping to uncover harmful content without requiring proactive data collection.
>
> Thus, this harmful data can accumulate naturally as vulnerabilities are discovered in the backbone, allowing SafePatching to serve as an effective **post-hoc remedy**.
>
> Our experimental results demonstrate that SafePatching significantly improves protection against both “Seen” and “Unseen” harmful data, illustrating its **generalization capabilities** applicable in real-world scenarios.
>
> Additionally, we have further validated SafePatching’s effectiveness in a more realistic setting of **Continual Post Safety Alignment** `(Section 5.3)`, showing its adaptability and robustness for ongoing deployment.
>
> Reference:
>
> [1] Wei A, Haghtalab N, Steinhardt J. Jailbroken: How does llm safety training fail?[J]. Advances in Neural Information Processing Systems, 2023, 36.
>
> [2] Bai Y, Jones A, Ndousse K, et al. Training a helpful and harmless assistant with reinforcement learning from human feedback[J]. arXiv preprint arXiv:2204.05862, 2022.
>
> [3] Perez E, Huang S, Song F, et al. Red Teaming Language Models with Language Models[C]//Proceedings of the 2022 Conference on Empirical Methods in Natural Language Processing. 2022: 3419-3448.

---

> ### Author Response · Authors · 2024-11-19
> **Rebuttal by Authors (2/2)**
>
> > Weakness 3: Stability of SafePatching Approach. SafePatching's dual-patch integration requires careful parameter tuning and may lack robustness or generalizability across different architectures or types of prompts.
>
> We would like to clarify that, in fact, we **have provided a detailed analysis of the robustness and stability for all hyperparameters** of SafePatching including overall retention rate $p$, top rate $a$ and $b$, and scale weight $\alpha$ and $\beta$ `(lines 513 - 516)`. Experimental results and detailed analysis on four backbone models (LLaMA-2, LLaMA-3, Gemma and Mistral) are shown `in Figures 5 through 9 in Appendix G.3`.
>
> We can draw two conclusions from the hyper-parameter analysis:
>
> - Random retention rate $p$ and the top rate $a$ and $b$ are **robust across different backbones**.
> - The only model-specific hyper-parameter we need to adjust is the scale weight $\alpha$ and $\beta$. This process involves merely assignment operations in storage space, making it **very quick and efficient**. It can be completed **without involving any GPU computations**.
>
> Regarding the generalizability across different types of prompts, we have evaluated SafePatching under Seen and Unseen harmful prompts from AdvBench `(Table 2)`, and different harmful categories from Beavertails dataset [1] under continual post-safety alignment `(Table 4 and Figure 4)`. Moreover, we supplement our experiments with additional evaluations on the Beavertails dataset, which introduces 14 different categories of harmful prompts.
>
> |LLaMA-2-7B-Chat|Beavertails|XSTest|OKTest|AVG.|MT-Bench|
> |--|--|--|--|--|--|
> |Original|19.64|8.00|4.67|40.35|6.01|
> |GA|3.65|35.20|38.67|39.02|4.01|
> |GA+Mismatch|3.98|22.40|25.33|38.42|4.56|
> |RESTA|5.39|66.33|41.60|39.39|5.49|
> |NPO|17.07|18.80|18.67|39.26|5.62|
> |SafeDecoding|**0.01**|80.80|59.67|/|5.72|
> |Self-CD|0.03|22.80|41.67|/|3.98|
> |ROSE|0.02|43.20|40.33|/|4.14|
> |SafePatching|7.87|**3.33**|**2.33**|**40.42**|**6.14**|
>
> These additional experiments demonstrate SafePatching’s ability to generalize across varied safety scenarios, reinforcing the method’s transferability and robustness.
>
> Reference:
>
> [1] Ji J, Liu M, Dai J, et al. Beavertails: Towards improved safety alignment of llm via a human-preference dataset[J]. Advances in Neural Information Processing Systems, 2023, 36.
>
> ---
>
> > Question 1: Could you please clarify and elaborate how they are implemented given a harmful dataset?
>
> **Gradient Ascent for the Safety Patch**
>
> The goal of the safety patch is to **reduce the likelihood of generating harmful outputs**. To achieve this, we perform gradient ascent on the loss associated with harmful responses. Specifically:
>
> - Objective: **Maximize the loss on harmful input-output pairs**. This effectively “teaches” the model to avoid or forget producing unsafe content by penalizing its unsafe responses.
>
> - Implementation: Please kindly refer to our code in supplementary files in the folder `src/GA/src/model/unlearned_model.py in line 14` to negate the sign of the loss.
>
> **Gradient Descent for the Over-Safety Patch**
>
> The over-safety patch ensures the model does not become overly cautious, which could lead to false refusals. Specifically:
>
> - Objective: **Minimize the loss on the same harmful input-output pairs**. This allows to remove the backbone’s internal safety defenses and enable it to respond freely to any input prompt.
>
> - Implementation: This is same with the standard SFT loss without any change.
>
> > Regarding Ethics Concern: Potential for Misuse and Safety Bypass
>
> Thank you for raising this important concern. As noted in Appendix I, we have discussed the potential risks associated with SafePatching. While the framework is designed to balance safety and usability by allowing benign responses to sensitive keywords, we recognize the potential risk of misuse by bad actors.
>
> To mitigate this, we:
> - Emphasize research-only usage, ensuring the framework is used solely for advancing safety research.
> - Conduct extensive evaluations (e.g., on Beavertails and AdvBench), demonstrating SafePatching improves safety without introducing significant bypass vulnerabilities.
> - Advocate for restricted access and transparency, ensuring controlled deployment and community oversight.
>
> We will further expand on these risks in the revised manuscript and plan to integrate dynamic adversarial testing to strengthen safety.
>
> ---
>
> We once again sincerely appreciate your comprehensive and valuable suggestions, which are crucial for enhancing the quality of our paper. We hope our response can alleviate concerns you may have regarding our paper and look forward to further communication with you.

---

> ### Author Response · Authors · 2024-12-02
> **Kind Reminder to Reviewer 1JaV**
>
> Dear Reviewer 1JaV,
>
> Could you please let us know if our responses satisfactorily address the issues you raised? We would greatly appreciate any further suggestions or clarifications you may have and are happy to discuss them further if needed.
>
> Thank you again for your time and consideration.

---

### Official Review · Reviewer_c4UP · 2024-11-04

**Soundness:** 2
**Presentation:** 3
**Contribution:** 2
**Rating:** 5
**Confidence:** 4

**Summary:**

This paper proposes a novel post-safety alignment (PSA) method, called SAFEPATCHING, which aims to address safety, over-safety, and utility issues in large language models (LLMs). In this paper, the authors develop a two-stage PSA framework, which applies distinct safety patches to the backbone LLM based on harmful data to improve safety and reduce over-safety, meanwhile, maintaining the utility capability of the LLM. The experiment shows that SAFEPATCHING achieves more effective and efficient PSA compared to baseline methods across four aligned LLMs

**Strengths:**

* The paper proposes a new method named SAFEPATCHING to address the limitations of existing methods on post-safety alignment for LLMs, such as over-safety issues and high cost.
* The paper presents experimental results and comparisons with state-of-the-art methods to demonstrate the effectiveness of SAFEPATCHING and uses multiple open-source datasets on safety, over-safety, and utility for a comprehensive evaluation. Besides, this paper has interesting findings on the distribution of the most important parameters for safety and over-safety, providing future research directions for the community.

**Weaknesses:**

* Lack of justification in Sec. 3.3 controllable patching. The authors may want to highlight the novelty of their tool and the rigor of their method. Currently, it appears that the approach relies on the SNIP score proposed by Lee et al., as well as model merging methods by Yu et al. and Hui et al., without a thorough explanation of the unique contributions or advancements made in this work.
* Although the authors conducted an excessive experiment to show the effectiveness of SAFEPATCHING, several concerns existed in the settings.
   * The study evaluates SAFEPATCHING using only a single harmful dataset, AdvBench, which may not adequately demonstrate the method's transferability across different safety scenarios. Given the extensive range of safety categories and perspectives, it's essential to assess whether a backbone LLM patched using AdvBench can maintain its effectiveness on other datasets representing diverse types of harmful content.
   * The authors did not specify how they fine-tuned the Longformer-based judger. Wang et al. used annotated data generated through human labor to fine-tune their Longformer model. It remains unclear whether the fine-tuned model from Wang et al.'s work was directly utilized in this experiment or if further adjustments were made. Clarification on this point would provide a better understanding of the model’s setup and any adaptations relevant to this study.

Yu, Le, et al. "Language models are super mario: Absorbing abilities from homologous models as a free lunch." Forty-first International Conference on Machine Learning. 2024.

Hui, Tingfeng, et al. "HFT: Half Fine-Tuning for Large Language Models."

Wang, Yuxia, et al. "Do-not-answer: A dataset for evaluating safeguards in llms."

**Questions:**

* Given that only the AdvBench dataset was used to evaluate SAFEPATCHING, how does the method perform across other safety-related datasets? Could testing with a broader range of harmful data enhance our understanding of its transferability to diverse safety scenarios?
* Since the authors did not specify whether they directly used the fine-tuned Longformer model from Wang et al. or performed additional fine-tuning, what impact might this setup have on the accuracy and reliability of the judgment model in this experiment?
* Could a deeper explanation of these aspects clarify the novelty and rigor of the proposed approach in Section 3.3?

---

> ### Author Response · Authors · 2024-11-19
> **Rebuttal by Authors (1/2)**
>
> We appreciate your positive feedback on our experimental results, comparisons with state-of-the-art methods, and insights into parameter distributions. We would address each of the concerns in detail:
>
> ---
>
> > Weakness 1 and Question 3: Could a deeper explanation of these aspects clarify the novelty and rigor of the proposed approach in Section 3.3?
>
> Our SafePatching introduces a novel approach by implementing a controllable dual-patching process that effectively `(results in Table 2 and 3)` and efficiently `(results in Table 1)` achieves the three primary objectives of post-safety alignment (PSA): enhancing safety, reducing over-safety, and maintaining utility.
>
> - Regarding SNIP: In previous works, SNIP has been primarily used to analyze models’ safety-related behaviors. By contrast, we employ SNIP to actively enhance post-alignment performance, extending its original application to a new context that focuses on achieving balanced PSA goals.
>
> - Regarding model merging methods by Yu et al. and Hui et al.: Rather than directly adopting their methods, we drew inspiration from their discussions on **parameter sparsification**. This guided us in designing our approach.
>
> Building on these ideas, we further propose **patching within the difference set of the most important parameter regions for each patch**, reducing conflicts between the safety and over-safety patches. As you noted, our analysis in Figure 3 provides interesting insights into the distribution of these key parameters, highlighting future research directions for the community.
>
> Also, our **experimental results in `Table 3` against state-of-the-art model merging methods** also demonstrate the effectiveness of our controllable dual-patching proposed in Section 3.3.
>
> ---
>
> > Weakness 2.1 and Question 1: how does the method perform across other safety-related datasets? Could testing with a broader range of harmful data enhance our understanding of its transferability to diverse safety scenarios?
>
> We agree that validating our method across diverse safety scenarios would strengthen our results.
>
> And we would like to clarity that we have evaluated SafePatching on 3 different harmful categories from Beavertails dataset [1] under continual post-safety alignment `(lines 520 – 528, Table 4 and Figure 4 in Section 5.3)`.
>
> To further address this, we supplement our experiments with additional evaluations on the Beavertails dataset, which introduces 14 different categories and perspectives of harmful content.
>
> |LLaMA-2-7B-Chat|Beavertails|XSTest|OKTest|AVG.|MT-Bench|
> |--|--|--|--|--|--|
> |Original|19.64|8.00|4.67|40.35|6.01|
> |GA|3.65|35.20|38.67|39.02|4.01|
> |GA+Mismatch|3.98|22.40|25.33|38.42|4.56|
> |RESTA|5.39|66.33|41.60|39.39|5.49|
> |NPO|17.07|18.80|18.67|39.26|5.62|
> |SafeDecoding|**0.01**|80.80|59.67|/|5.72|
> |Self-CD|0.03|22.80|41.67|/|3.98|
> |ROSE|0.02|43.20|40.33|/|4.14|
> |SafePatching|7.87|**3.33**|**2.33**|**40.42**|**6.14**|
>
> These additional experiments demonstrate SafePatching’s ability to generalize across varied safety scenarios, reinforcing the method’s transferability and robustness beyond AdvBench alone. We will include these results in the revised paper to offer a more comprehensive evaluation of SafePatching’s effectiveness across multiple safety domains.
>
> Reference:
>
> [1] Ji J, Liu M, Dai J, et al. Beavertails: Towards improved safety alignment of llm via a human-preference dataset[J]. Advances in Neural Information Processing Systems, 2023, 36.

---

> ### Author Response · Authors · 2024-11-19
> **Rebuttal by Authors (2/2)**
>
> > Weakness 2.2 and Question 2: The authors did not specify how they fine-tuned the Longformer-based judger. What impact might this setup have on the accuracy and reliability of the judgment model in this experiment?
>
> We apologize for any ambiguity. In our experiment, we **directly adopt** the Longformer model provided by Wang et al. **without any further finetuning**.
>
> To validate the accuracy and reliability of the judgment model used in our experiments, we sample 50 instances from AdvBench and compare its classification performance with GPT-4o, where human judgements serve as the golden reference. This comparison confirms the Longformer-based model’s effectiveness in our specific setting.
>
> |Model|Accuracy|
> |---------------------|----------|
> |Longformer|96.70%|
> |GPT-4o|97.80%|
>
> > Regarding Ethics Concern: Potential for Misuse and Safety Bypass
>
> Thank you for raising this important concern. As noted in Appendix I, we have discussed the potential risks associated with SafePatching. While the framework is designed to balance safety and usability by allowing benign responses to sensitive keywords, we recognize the potential risk of misuse by bad actors.
>
> To mitigate this, we:
> - Emphasize research-only usage, ensuring the framework is used solely for advancing safety research.
> - Conduct extensive evaluations (e.g., on Beavertails and AdvBench), demonstrating SafePatching improves safety without introducing significant bypass vulnerabilities.
> - Advocate for restricted access and transparency, ensuring controlled deployment and community oversight.
>
> We will further expand on these risks in the revised manuscript and plan to integrate dynamic adversarial testing to strengthen safety.
>
> ---
>
> We once again sincerely appreciate your comprehensive and valuable suggestions, which are crucial for enhancing the quality of our paper. We hope our response can alleviate concerns you may have regarding our paper and look forward to further communication with you.

---

> ### Author Response · Authors · 2024-12-02
> **Kind Reminder to Reviewer c4UP**
>
> Dear Reviewer c4UP,
>
> Could you please let us know if our responses satisfactorily address the issues you raised? We would greatly appreciate any further suggestions or clarifications you may have and are happy to discuss them further if needed.
>
> Thank you again for your time and consideration.

---

### Official Review · Reviewer_mxyn · 2024-11-04

**Soundness:** 3
**Presentation:** 3
**Contribution:** 3
**Rating:** 6
**Confidence:** 4

**Summary:**

This paper proposes a post safety alignment method which merges two models post-trained on harmful data with gradient ascent and descent respectively. The post-trained and merged model preserves a balance on safety, over-safety mitigation, and utility preservation.

**Strengths:**

•	The idea is straightforward.
•	The experiments are extensive.

**Weaknesses:**

•	The paper lacks the comparision of external safeguards methods such as OpenChatKit and NeMo guardrails that are known to handle over-safety issues. Would these external safeguards methods also achieve the three objectives proposed in the paper?
•	There are a few hyperparameters in equation 7&8, such as a, b, \alpha, \beta. How you set these parameters? In Table 3, merging methods like the task arithmetic and TIES-merging do not have big differences compared to the intersect patch. Would the benefit comes from your hyperparameter selection?

**Questions:**

Would you please address the concerns in weakness?

---

> ### Author Response · Authors · 2024-11-19
> **Rebuttal by Authors**
>
> Thank you for your constructive feedback and we address your concerns as follows.
>
> ---
>
> > Weakness 1: Would these external safeguards methods, OpenChatKit and NeMo, also achieve the three objectives proposed in the paper?
>
> Thank you for your valuable suggestion to compare our method with external safeguard methods like OpenChatKit and NeMo guardrails. The **results on four backbones, shown in the tables below**, indicate that while these external methods meet the objective of utility preservation, they could not solve the conflict between safety and over-safety to achieve comprehensive PSA. By contrast, our SafePatching still demonstrates strong advantages.
>
> |**LLaMA-2-7B-Chat**|Seen|Unseen|XSTest|OKTest|MT-Bench|
> |--|--|--|--|--|--|
> |Original|24.00|21.00|8.00|4.67|6.01|
> |NeMo|**6.00**|8.50|74.00|86.33|5.78|
> |OpenChatKit|24.00|21.00|10.00|6.00|6.04|
> |SafePatching|7.25|**7.50**|**3.33**|**2.33**|**6.14**|
>
> |**LLaMA-3-8B-Instruct**|Seen|Unseen|XSTest|OKTest|MT-Bench|
> |--|--|--|--|--|--|
> |Original|12.50|18.50|2.80|9.67|8.20|
> |NeMo|**1.00**|**1.00**|15.20|14.67|8.11|
> |OpenChatKit|12.50|18.50|4.40|11.00|8.17|
> |SafePatching|4.75|13.50|**1.60**|**6.33**|**8.18**|
>
> |Gemma-1.1-7B-it|Seen|Unseen|XSTest|OKTest|MT-Bench|
> |--|--|--|--|--|--|
> |Original|24.50|34.00|28.80|23.33|6.82|
> |NeMo|**1.50**|**1.00**|36.00|30.00|6.74|
> |OpenChatKit|24.50|34.00|28.80|23.33|6.83|
> |SafePatching|1.75|15.00|**16.00**|**14.67**|**6.94**|
>
> |Mistral-7B-Instruct-v0.1|Seen|Unseen|XSTest|OKTest|MT-Bench|
> |--|--|--|--|--|--|
> |Original|75.00|69.5|14.00|6.33|6.49|
> |NeMo|26.25|25.50|21.00|6.33|**6.45**|
> |OpenChatKit|75.00|69.50|15.00|6.33|6.43|
> |SafePatching|**6.75**|**15.50**|**5.20**|**3.33**|6.38|
>
> ---
>
> > Weakness 2: There are a few hyperparameters in equation 7&8, such as a, b, \alpha, \beta. How you set these parameters?
>
> We would like to clarify that, in fact, **we have provided** a detailed analysis of the robustness and selection process for these hyperparameters including overall retention rate $p$, top rate $a$ and $b$, and scale weight $\alpha$ and $\beta$ `(lines 513 - 516)`. Experimental results on four backbone models (LLaMA-2, LLaMA-3, Gemma and Mistral) are shown `in Figures 5 through 9 in Appendix G.3`, demonstrating the impact of different settings and the rationale behind our final choices.
>
> We can draw two conclusions from the hyper-parameter analysis:
>
> - Random retention rate $p$ and the top rate $a$ and $b$ are robust across different backbones.
> - The only model-specific hyper-parameter we need to adjust is the scale weight $\alpha$ and $\beta$, and:
>   - Larger $\alpha$ and $\beta$ can negatively impact overall model performance and lead to meaningless responses, potentially leading to a loss of utility (especially on Mistral and Gemma).
>   - Within an appropriate value range, $\alpha$ and $\beta$ further balance the conflict between safety and over-safety: the larger $\alpha$, the more pronounced the effect of the Safety Patch, enhancing safety performance but increasing over-safety. Conversely, a larger $\beta$ has the opposite effect.
>
> ---
>
> > Weakness 3: Would the benefit in arithmetic and TIES-merging comes from your hyperparameter selection?
>
> We would like to clarify that the results for Arithmetic and TIES-merging **have already been obtained after optimal hyperparameter tuning for each method**. For further clarity, we report the hyperparameter tuning process for them in terms of $\alpha$ and $\beta$, similar to that process used for our method, in the table below. Best results (reported in current paper) are in bold and ‘/’ means the ability of resulting model is largely damaged to get meaningful outputs.
>
> |TaskArithmeticetic|Seen|Unseen|XSTest|OKTTest|AVG.|MT-Bench|
> |--|--|--|--|--|--|--|
> |(1.0,0.05)|/|/|/|/|/|/|
> |(1.0,0.1)|14.00|12.00|12.00|15.00|39.96|5.99|
> |**(1.0,0.2)**|**7.50**|**7.50**|**7.33**|**8.40**|**40.34**|**6.08**|
> |(1.0,0.5)|26.75|25.00|3.20|6.33|40.21|6.03|
> |(1.0,1.0)|56.25|57.75|1.60|1.33|39.99|5.93|
> |(1.0,1.5)|85.25|87|0.00|0.00|39.58|5.94|
>
> |TIES-Merge|Seen|Unseen|XSTest|OKTTest|AVG.|MT-Bench|
> |--|--|--|--|--|--|--|
> |(1.0,0.05)|/|/|/|/|/|/|
> |(1.0,0.1)|4.50|3.00|18.40|24.67|39.77|5.96|
> |**(1.0,0.2)**|**12.00**|**11.00**|**7.67**|**6.00**|**40.43**|**6.09**|
> |(1.0,0.5)|25.75|27.50|3.20|5.00|40.21|6.03|
> |(1.0,1.0)|61.25|65.50|1.20|1.33|39.99|5.93|
> |(1.0,1.5)|85.00|85.50|0.00|0.00|39.58|5.94|
>
> These results demonstrate that our SafePatching still maintains an advantage in performance across various scale weights used for Arithmetic and TIES-merging, highlighting its robustness in achieving a balanced trade-off between safety, over-safety mitigation, and utility preservation.
>
> ---
>
> We once again sincerely appreciate your comprehensive and valuable suggestions, which are crucial for enhancing the quality of our paper. We hope our response can alleviate concerns you may have regarding our paper and look forward to further communication with you.

---

> ### Author Response · Authors · 2024-12-02
> **Kind Reminder to Reviewer mxyn**
>
> Dear Reviewer mxyn,
>
> Could you please let us know if our responses satisfactorily address the issues you raised? We would greatly appreciate any further suggestions or clarifications you may have and are happy to discuss them further if needed.
>
> Thank you again for your time and consideration.

---

### Official Review · Reviewer_pyqz · 2024-11-06

**Soundness:** 2
**Presentation:** 2
**Contribution:** 2
**Rating:** 3
**Confidence:** 3

**Summary:**

The paper presents a method called SafePatching for post safety alignment (PSA) of large language models (LLMs). The authors claim that SafePatching addresses three PSA objectives: safety enhancement, over-safety mitigation, and utility preservation.

**Strengths:**

1. The problem addressed—post-hoc safety alignment—is important for ensuring that LLMs behave safely in real-world applications.
2. The empirical evaluation and ablations are fairly comprehensive across different LLM backbones and benchmarks.
3. The method shows some promise in balancing safety with utility preservation compared to existing baselines.

**Weaknesses:**

1. The proposed approach seems to be largely composed of a series of straightforward adaptations or incremental improvements on recent work. For instance, the use of gradient ascent and descent techniques for deriving safety and over-safety patches is largely an adaptation of existing machine unlearning methods described in the paper, rather than a truly novel contribution. The concept of patching the difference set of important parameters between safety and over-safety patches is perhaps the most novel aspect. However, it's still a relatively straightforward extension of existing ideas in parameter importance and model merging.
2. While the proposed approach does demonstrate that it is the only one to improve safety, over-safety, and utility over the backbone, in many cases, it performs significantly worse than the baselines for a particular safety or over-safety benchmark. Moreover, the safety and over-safety improvements over the backbone model are quite marginal in some cases. This highlights that there is more work to be done in effectively controlling the balance between safety enhancement and over-safety mitigation than the approach in its current state.

**Questions:**

1. How does the use of gradient ascent and descent for patch derivation differ from recent work in unlearning?

---

> ### Author Response · Authors · 2024-11-19
> **Rebuttal by Authors (2/2)**
>
> > Weakness 2.3: This highlights that there is more work to be done in effectively controlling the balance between safety enhancement and over-safety mitigation than the approach in its current state.
>
> We appreciate your insights and agree that balancing these objectives is a challenging but crucial aspect of PSA.
>
> - Through our experiments, we first identity this crucial challenge and demonstrate the **clear limitations of existing PSA methods in this regard**.
>
> - As detailed above, SafePatching is currently **the most effective and efficient method** for simultaneously achieving comprehensive PSA, representing a **pioneering exploration in this area**.
>
> - While we totally agree with you that there is still room for further improvement, we hope this work inspires the community to continue advancing the field.
>
> ---
>
> > Question 1: How does the use of gradient ascent and descent for patch derivation differ from recent work in unlearning?
>
> Thank you for your question. We directly adopt gradient ascent and descent techniques from recent work in unlearning. However, they are **only a small component of SafePatching**, specifically used to generate the safety and over-safety patches. And proposing new unlearning or fine-tuning techniques lies outside the scope of our study. Instead, the focus of SafePatching is to provide **an efficient `(results in Table 1)` and effective `(results in Table 2 and 3)` solution** for achieving the three PSA objectives—safety enhancement, over-safety mitigation, and utility preservation—simultaneously.
>
> We believe the interesting part in gradient ascent and descent is that we leverage **a single harmful dataset to generate both types of patches**, thereby avoiding additional data overhead. This efficient use of resources may enhance the framework’s practical utility in real-world applications.
>
> ---
>
> We once again sincerely appreciate your comprehensive and valuable suggestions, which are crucial for enhancing the quality of our paper. We hope our response can alleviate concerns you may have regarding our paper and look forward to further communication with you.

---

> ### Author Response · Authors · 2024-11-19
> **Rebuttal by Authors (1/2)**
>
> We appreciate your acknowledgment of our comprehensive empirical evaluations across different LLM backbones and benchmarks. Additionally, we are glad to see that you find merit in our approach’s ability to balance safety with utility preservation, as compared to existing baselines. We will address your concerns in detail as follows.
>
> ---
>
> > Weakness 1: the use of gradient ascent and descent techniques is not a truly novel contribution. The concept of patching the difference set of important parameters is still a relatively straightforward extension of existing ideas in parameter importance and model merging.
>
> We thank you for your insights and the opportunity to clarify our core contribution.
>
> **Regarding gradient ascent and descent techniques**
>
> - Gradient ascent and descent techniques are indeed part of our framework, but they are **only a small component of SafePatching**, specifically used to generate the safety and over-safety patches. And proposing new unlearning or fine-tuning techniques lies outside the scope of our study. We did not intend to position these techniques as a primary contribution of our work. Instead, the focus of SafePatching is to provide **an efficient (`results in Table 1`) and effective (`results in Table 2 and 3`) solution** for achieving the three PSA objectives.
>
> - In addition, we believe the interesting part in gradient ascent and descent is that we leverage **a single harmful dataset to generate both types of patches**, thereby avoiding additional data overhead. This efficient use of resources may enhance the framework’s practical utility in real-world applications.
>
> **Regarding patching the difference set of important parameters (controllable dual-patching)**
>
> - For parameter importance score SNIP: In previous works, SNIP has been primarily used to analyze models’ safety-related behaviors. By contrast, we employ SNIP to actively enhance post-alignment performance, extending its original application to a new context that focuses on achieving balanced PSA goals.
>
> - For model merging methods: We propose controllable dual-patching to **patch within the difference set of the most important parameter regions for each patch**, reducing conflicts between the safety and over-safety patches. Our experimental results in `Table 3` against state-of-the-art model merging methods demonstrate the effectiveness of our controllable dual-patching. And our analysis in `Figure 3` provides interesting insights into the distribution of these key parameters. Thus, both our experimental results and analytical studies demonstrate that SafePatching consistently outperforms direct applications of current model merging techniques in addressing PSA objectives.
>
> ---
>
> > Weakness 2.1: The proposed method performs significantly worse than the baselines for a particular safety or over-safety benchmark.
>
> We thank you for pointing this observation and would like to clarify the strengths of SafePatching as demonstrated in our experimental results. Specifically:
>
> - As shown in `Tables 2, 9, and 10`, across **all four backbone models**, our SafePatching achieves **top 3 performance** in safety and **top 2 performance** in over-safety consistently.
>
> - In contrast, baselines that **achieve the best performance in either safety or over-safety tend to severely degrade the other aspect**, often causing significant trade-offs. For example, almost all current PSA methods optimized solely for safety, often leading to severe over-safety violations.
>
> These results highlight that **SafePatching achieves Pareto-optimal performance** and strikes a more balanced trade-off, making it uniquely effective for addressing all three PSA objectives simultaneously.
>
> ---
>
> > Weakness 2.2: Moreover, the safety and over-safety improvements over the backbone model are quite marginal in some cases.
>
> We appreciate you feedback and would like to clarify that the improvements achieved by our method in terms of safety and over-safety are far from marginal.
>
> - Safety: SafePatching demonstrates significant improvements in safety performance across all backbones:
>   - On LLaMA-2, safety improves by **67.22%**,
>   - On LLaMA-3 by **41.13%**,
>   - On Gemma by **71.37%**,
>   - And on Mistral by **84.60%**.
>
> - Over-Safety: Our method also achieves notable reductions in over-safety, with improvements of:
>   - **55.33%** on LLaMA-2,
>   - **36.41%** on LLaMA-3,
>   - **41.17%** on Gemma,
>   - And **58.04%** on Mistral.
>
> Crucially, SafePatching is among the very few methods that effectively mitigate over-safety. **Almost no other baselines achieve any reduction in over-safety**; instead, they tend to significantly exacerbate the problem. For example, SafeDecoding leads to an over-safety rate increase from **8% to 80.8%** on the XSTest benchmark with LLaMA-2, demonstrating its failure to balance safety and over-safety.
>
> Thank you for your valuable comments. We will revise the manuscript to better highlight these gains.

---

### Official Review · Reviewer_Kpns · 2024-11-23

**Soundness:** 3
**Presentation:** 3
**Contribution:** 3
**Rating:** 5
**Confidence:** 3

**Summary:**

This paper investigates how to balance safety enhancement and over-safety mitigation while retaining model utility during post-safety alignment. To achieve this, the authors propose the SAFEPATCHING framework, which optimizes separate patches for safety enhancement and over-safety mitigation. Through controllable patching, these two patches are selectively merged at the parameter level, addressing conflicts between them while preserving model utility. The paper includes a comprehensive evaluation across three dimensions—safety enhancement, over-safety mitigation, and utility preservation—demonstrating the effectiveness of SAFEPATCHING. Additionally, it provides a deeper insight by analyzing parameter selection, the distribution of each patch's parameters, and other aspects.

**Strengths:**

- This paper is well-structured and easy to follow.

- The motivation of the paper is clear, and the novel method proposed aligns well with this motivation.

- The experiments are thorough, providing a comprehensive evaluation across the three objectives on various baseline models and methods. Additionally, detailed analyses are conducted throughout.

**Weaknesses:**

- The **experimental setting** section seems to overlook the choice of hyperparameters. It appears that key hyperparameters like top rate, scale weight, etc., are only mentioned in the appendix, with no indication in the main text of how these crucial settings were determined for the primary experiments (if I missed this, please let me know).

- **Minor suggestions and areas for improvement** (though not sufficient reasons for rejection): The experimental section is somewhat dense, especially section 5.2. Breaking it down into more subsections or adjusting the layout could enhance readability. Important hyperparameters like top rate, scale weight, and retention rate should ideally be summarized in a table or have a representative results diagram in the main text rather than placing all analysis in the appendix.

**Questions:**

- The results for top rate were somewhat surprising, as the model appears overly robust to variations in top rate. A small question arises here: could this be due to the narrow range of top rate choices? Expanding the range to include smaller or larger top rates might yield more insights.

- **Figure 3 Insight**: The distribution of overly-safety and safety parameters across Transformer layers shown in Figure 3 is very intriguing, especially with one concentrating in the middle layers and the other in the lower layers. Could you provide any insights or explanations for this phenomenon?

---

> ### Author Response · Authors · 2024-11-25
> **Rebuttal by Authors (1/2)**
>
> We thank you for your thoughtful and constructive feedback on our paper. We appreciate your recognition of the novelty and clarity of our method, as well as the comprehensive evaluation we conducted. Below, we address each of the weaknesses you raised.
>
> ---
>
> > Weakness 1: The experimental setting section seems to overlook the choice of hyperparameters.
>
> Thank you for pointing out this potential oversight. In the initial version of the paper, we included the hyperparameter settings for our main experiments in `Table 6 (page 18 in the Appendix)`. However, we now realize that this placement might lead to readers overlooking this critical information. Based on your suggestion, we have **moved these details to `Table 1 (page 7 in the Main Text)` in the revised version** to ensure better accessibility and clarity.
>
> We greatly appreciate your feedback in helping us improve the presentation of our work.
>
> ---
>
> > Weakness 2: The experimental section is somewhat dense, especially section 5.2.
>
> Thank you for your valuable feedback. We have revised the experimental section in the manuscript following your suggestions. Changes are highlighted in **orange font** for clarity. Specifically:
>
> - We have moved the main hyperparameter settings from the appendix to **Table 1 on page 7** in the main text.
>
> - The primary experimental results on LLaMA-3 and training/inference efficiency analysis have been relocated to the appendix to streamline the main text.
>
> - Section 5.2 has been divided into three subsections: **Ablation Study**, **Distribution of Important Parameters**, and **Robustness Analysis on Hyper-Parameters**. Additionally, we modified the first sentence of each paragraph to clearly convey the main conclusion of the respective analysis.
>
> These changes aim to improve the readability and organization of the experimental section, ensuring that readers can navigate the content more effectively. We sincerely appreciate your feedback, which has greatly contributed to enhancing the presentation of our work.
>
> ---
>
> > Question 1: could the robustness of top rate be due to the narrow range of top rate choices? Expanding the range to include smaller or larger top rates might yield more insights.
>
> Thank you for your insightful question. We have conducted additional experiments to investigate the impact of both smaller and larger top rates on LLaMA-2-7B-Chat (top rate values constrained within 30%, as it cannot exceed the overall retention rate $p$).
>
> |Top Rate (%)|Seen|Unseen|XSTest|OKTest|AVG.|MT-Bench|
> |-|-|-|-|-|-|-|
> |(0.1, 0.1)|15|16|8.00|4.67|40.23|6.02|
> |(0. 5, 0. 5)|10|10|6.40|4.33|40.37|6.01|
> |(10, 10)|7.25|8|4.00|2.33|40.03|5.98|
> |(15, 15)|7.75|8.25|4.40|3.33|39.79|5.91|
> |(20, 20)|8.25|9.25|5.20|4.33|39.72|5.90|
> |(25, 25)|8.25|8.75|5.20|3.33|39.88|5.93|
>
> The results show that while the model remains robust across all three evaluation dimensions with larger top rates, smaller top rates lead to performance degradation. This can be explained as follows:
>
> - Larger top rates (e.g., 25%): Even with higher top rates, the parameters in the difference set account for only about 10% of the total parameters, maintaining a **highly sparse** update. This aligns with findings in recent studies showing that even a small fraction of parameters can have a significant impact on large model performance [1,2].
>
> - Smaller top rates: With smaller top rates, the difference set **contains almost no parameters** from either the Safety or Over-Safety Patch, making it difficult to leverage their respective contributions effectively.
>
> We appreciate your suggestion, as it has helped us provide a more comprehensive analysis of the impact of top rate variations, further strengthening the paper. Thank you again for your valuable feedback!
>
> Reference:
>
> [1] Yu L, Yu B, Yu H, et al. Language models are super mario: Absorbing abilities from homologous models as a free lunch[C]. ICML 2024.
>
> [2] Wei B, Huang K, Huang Y, et al. Assessing the Brittleness of Safety Alignment via Pruning and Low-Rank Modifications[C]. ICML 2024.

---

> ### Author Response · Authors · 2024-11-25
> **Rebuttal by Authors (2/2)**
>
> > Question 2: Could you provide any insights or explanations for the phenomenon in Figure 3?
>
> Thank you for highlighting this interesting observation. Regrettably, our current conclusions are empirical, and we cannot yet provide definitive explanations. However, our findings are consistent with recent work in the field, which offers some related insights:
>
> - Over-Safety Parameters: These parameters are identified through gradient descent on unsafe data, representing the model’s unsafe parameter set. A recent study [1] also found that unsafe parameters are predominantly located in the lower layers of the model. Their experiments showed that adding the representations of lower layers to the model’s final output significantly weakens its safety performance.
>
> - Safety Parameters: Similarly, recent research [2] has observed that parameters associated with safety performance tend to concentrate in the middle layers of Transformer models. This aligns closely with our empirical findings.
>
> We appreciate your question and agree that this is a promising direction for future exploration. To that end, we plan to investigate this phenomenon further using Tulu 3 [3], a model that was open-sourced just two days ago and is the first to provide full transparency in its safety post-training process. This transparency offers a unique opportunity to deepen our understanding of how safety mechanisms are formed within models.
>
> Thank you again for your insightful question and for motivating us to pursue this exciting research avenue!
>
> Reference:
>
> [1] Ghandeharioun A, Yuan A, Guerard M, et al. Who's asking? User personas and the mechanics of latent misalignment[J]. arXiv preprint arXiv:2406.12094, 2024.
>
> [2] Li S, Yao L, Zhang L, et al. Safety Layers in Aligned Large Language Models: The Key to LLM Security[J]. arXiv preprint arXiv:2408.17003, 2024.
>
> [3] Lambert N, Morrison J, Pyatkin V, et al. TÜLU 3: Pushing Frontiers in Open Language Model Post-Training[J].
>
> ---
>
> We once again sincerely appreciate your comprehensive and valuable suggestions, which are crucial for enhancing the quality of our paper. We hope our response can alleviate concerns you may have regarding our paper and look forward to further communication with you.

---

> ### Author Response · Authors · 2024-11-28
> **Kind Reminder to Reviewer Kpns**
>
> Dear Reviewer Kpns,
>
> Could you please let us know if our responses satisfactorily address the issues you raised? We would greatly appreciate any further suggestions or clarifications you may have and are happy to discuss them further if needed.
>
> Thank you again for your time and consideration.

---

### Author Response · Authors · 2024-11-26
**Kind Reminder to All Reviewers**

Dear All Reviewers,

Could you please let us know if our responses satisfactorily address the issues you raised? We would greatly appreciate any further suggestions or clarifications you may have and are happy to discuss them further if needed.

Thank you again for your time and consideration.

---

### Note · Authors · 2024-12-10

I have read and agree with the venue's withdrawal policy on behalf of myself and my co-authors.